# Non-Contact Fall Detection System Using 4D Imaging Radar for Elderly Safety Based on a CNN Model

**DOI:** 10.3390/s25113452

**Published:** 2025-05-30

**Authors:** Sejong Ahn, Museong Choi, Jongjin Lee, Jinseok Kim, Sungtaek Chung

**Affiliations:** 1Department of Computer Engineering, Tech University of Korea, Siheung 15073, Republic of Korea; toyou85@tukorea.ac.kr (S.A.); gttkim@hanmail.net (J.K.); 2Department of Bio Health Convergence Engineering, Tech University of Korea, Siheung 15073, Republic of Korea; 00tyty003@tukorea.ac.kr (M.C.); gggg9806@tukorea.ac.kr (J.L.)

**Keywords:** fall detection, 4D imaging radar sensor, CNN, point cloud, elderly safety

## Abstract

Progressive global aging has increased the number of elderly individuals living alone. The consequent rise in fall accidents has worsened physical injuries, reduced the quality of life, and increased medical expenses. Existing wearable fall-detection devices may cause discomfort, and camera-based systems raise privacy concerns. Here, we propose a non-contact fall-detection system that integrates 4D imaging radar sensors with artificial intelligence (AI) technology to detect falls through real-time monitoring and visualization using a web-based dashboard and Unity engine-based avatar, along with immediate alerts. The system eliminates the need for uncomfortable wearable devices and mitigates the privacy issues associated with cameras. The radar sensors generate Point Cloud data (the spatial coordinates, velocity, Doppler power, and time), which allow analysis of the body position and movement. A CNN model classifies postures into standing, sitting, and lying, while changes in the speed and position distinguish falling actions from lying-down actions. The Point Cloud data were normalized and organized using zero padding and k-means clustering to improve the learning efficiency. The model achieved 98.66% accuracy in posture classification and 95% in fall detection. This study demonstrates the effectiveness of the proposed fall detection approach and suggests future directions in multi-sensor integration for indoor applications.

## 1. Introduction

The rapidly aging population as a result of declining birth rates and advancements in medical technology has led to an increase in the number of single-person elderly households. These demographic changes significantly impact the economy, social welfare, housing conditions, and labor markets and necessitate new policy and technological approaches. Particularly, factors such as spousal loss, children gaining their independence, and the pursuit of autonomy contribute to the rise in single-person elderly households, which, while enabling independent lifestyles, also raise concerns about vulnerability during emergencies [1].

Health-related conditions such as sarcopenia, impaired balance, and cognitive decline are widely recognized as major contributors to falls experienced by older adults. Fhon et al. [2] conducted a systematic review in which they highlighted the strong association between sarcopenia, the risk of falling, and cognitive impairment. Similarly, Oytun et al. [3] examined the relationship between the risk of falling and frailty, sarcopenia, and balance disturbances in individuals with mild-to-moderate Alzheimer’s disease. Chen et al. [4] further supported these findings by demonstrating the link between sarcopenia and cognitive dysfunction through meta-analysis. In addition, Kim and Kim [5] emphasized that stress and depression significantly affect the quality of life in single-person elderly households and found these factors to further aggravate the vulnerability of these persons to health-related issues.

The growing elderly population has increased the demand for technological interventions that support safe and independent aging. Rubenstein [6] maintained that fall prevention is essential to sustaining the mobility and independence of older adults, while Wang et al. [7] reviewed a wide range of smart fall detection systems and highlighted their potential to improve safety and reduce healthcare costs through timely intervention and monitoring.

Traditional fall-detection methods are generally divided into contact-based and non-contact-based approaches. Previous studies have primarily focused on contact-based methods, such as equipping wearable devices with Inertial Measurement Unit (IMU) sensors—typically three-axis accelerometers—worn on the wrist or waist to detect falls. Xefteris et al. [8] and Villa and Casilari [9] reviewed various wearable-based fall detection systems and highlighted concerns regarding their long-term use, such as physical discomfort and psychological resistance among elderly users who are required to wear the device on an ongoing basis. Additionally, wearable devices rely on wireless communication (e.g., Bluetooth or Wi-Fi) and are battery-powered, which is a limitation for long-term monitoring.

As a non-contact alternative, camera-based fall-detection systems have been proposed. These systems involve the installation of cameras on ceilings or walls in indoor environments with the intention of addressing the aforementioned issues related to wearing discomfort and battery charging. However, camera-based approaches often perform poorly in low-light environments (e.g., at night) and continue to raise serious concerns with respect to privacy violations due to their placement in personal living spaces. De Miguel et al. [10] discussed the trade-offs between monitoring effectiveness and privacy, whereas Ren and Peng [11] introduced image transmission techniques that activate only during suspected fall events. Koshmak et al. [12] also proposed facial blurring as a means of protecting user identity, though these methods do not fundamentally eliminate the risks of privacy infringement, which remains a contentious issue in image-based monitoring.

Our attempts to overcome these limitations led us to propose a non-contact fall-detection system that integrates 4D imaging radar sensors with artificial intelligence (AI) technology. The proposed system aims to effectively detect fall incidents among the elderly by monitoring the user’s current status and visualizing their status in real time through a Unity-based avatar. The radar sensors emit high-frequency signals and analyze the reflected signals to detect the position, velocity, and movement of the target. Because the system operates without physical contact, it eliminates wearing discomfort and obviates the need to charge the battery, as would be required for conventional contact-based methods. Moreover, because 4D radar generates Point Cloud data rather than video images, the user’s privacy is not infringed, and stable target detection is possible even in poorly lit environments (e.g., during nighttime or in low-light environments) [13].

In this study, data normalization, zero padding, and k-means clustering were applied as preprocessing techniques to classify three postures—standing, sitting, and lying—based on Point Cloud data. Additionally, the spatial invariance property of convolutional neural networks (CNNs) was utilized to effectively classify postures observed from various angles and in different positions [14]. To further minimize confusion between the movements associated with lying down slowly and actually falling, an additional method was developed to calculate changes in both the position and acceleration. The intuitive monitoring advantages provided by conventional camera-based fall-detection systems were retained by designing an avatar-based monitoring system using the Unity engine to allow users to intuitively understand the state of the detected subject.

Ultimately, we developed a new type of fall detection and monitoring system by integrating 4D radar sensors with an AI-based analysis model and a real-time avatar visualization interface that contributes to improving the safety and quality of life for the elderly.

This paper is organized as follows: Section 2 reviews related studies on fall detection technologies. Section 3 presents the architecture of the proposed system, including radar data processing and AI modeling. Section 4 outlines the experimental setup and presents the evaluation results. Section 5 discusses the implications, limitations, and directions for future research. Finally, Section 6 concludes this study.

## 2. Related Studies

### 2.1. Overview of Radar-Based Fall Detection

Radar-based sensing technologies have recently gained attention as promising solutions for fall detection and human activity monitoring, especially in contexts where privacy protection and non-contact operation are critical. Unlike wearable systems, which need to be worn on the body continuously, or camera-based approaches that raise privacy concerns, radar sensors offer the unique advantage of being able to detect motion and postures through walls and furniture, and in low-light environments, without capturing images that would enable the subject to be identified. These characteristics make radar sensing particularly well-suited for elderly monitoring in private residential settings.

### 2.2. Existing Radar-Based Posture Classification Studies

Recent studies on radar-based human posture recognition have generally adopted one of the following two approaches: (1) direct classification of the posture state (standing, sitting, or lying) or (2) the reconstruction of skeletal structures by estimating the positions of joints. The former approach is typically more computationally efficient and thus more suitable for real-time applications.

Baird et al. (2018) used ultra-wideband radar and a decision tree classifier to distinguish among the three postures—standing, sitting, or lying—and achieved an average accuracy of 84.94% [15]. Shrestha et al. (2020) employed a Bi-LSTM network on micro-Doppler signals from FMCW radar to classify six activities, including falling, with 91% accuracy [16]. Liang et al. (2021) proposed a Bi-LSTM-based model using 24 GHz radar SoC that achieved 99.25% accuracy for classifying four postures [17]. Their model has low computational complexity and is thus suitable for embedded systems.

Zhang et al. (2023) evaluated multiple machine learning models—including KNN and MLP—by processing mmWave radar-based Point Cloud data for six posture types and reported a peak accuracy of 94% with the MLP classifier [18]. Although their method was not radar-based, Werghi et al. (2002) demonstrated posture classification using 3D body scanner data and wavelet transform coefficients with accuracy as high as 98% across 19 posture classes [19]. Meng et al. (2020) proposed a deep neural architecture named mmGaitNet for identifying individuals based on radar-extracted gait features and reported an accuracy of more than 88% even in multi-subject environments [20].

### 2.3. Existing Radar-Based Fall Detection Studies

A variety of radar-based techniques for fall detection have also been proposed in recent years. Zhang et al. (2024) combined CNN and Bi-LSTM networks to process Doppler radar signals, which enabled them to achieve a fall detection accuracy of 98.83% [21]. Liu et al. (2023) developed a four-stage hierarchical algorithm using low-cost continuous-wave Doppler radar, which achieved 93.24% accuracy while also tracking the post-fall respiration rate [22]. Wu et al. (2023) utilized a time-frequency feature extraction method based on the Smoothed Pseudo Wigner–Ville Distribution (SPWVD) and applied XGBoost to reach 87.47% accuracy [23]. Other approaches include unsupervised anomaly detection with mmWave radar and real-time fall prediction using DBSCAN-based state transition modeling [24], as well as hybrid autoencoder-based systems such as mmFall that leverage variational RNNs for anomaly detection [25]. These studies highlight both the diversity of radar-based fall detection strategies and the potential for high accuracy under real-world conditions.

### 2.4. Contribution and Differentiation of This Study

Aiming to address the challenges presented by the abovementioned techniques, we proposed a lightweight yet accurate CNN-based architecture that directly classifies posture using preprocessed Point Cloud data obtained from a 4D radar sensor. The proposed system, which does not rely on joint estimation or Doppler domain features, achieved an average accuracy of 98.66% for classifying three postures—standing, sitting, and lying—using 5-fold cross-validation. This level of performance is either competitive with or exceeds that of prior studies while maintaining computational efficiency for real-time implementation.

A major distinguishing feature of the method developed in this study is the implementation of a Unity-based virtual environment for avatar visualization. This approach enables intuitive real-time monitoring, similar to camera-based systems, yet it circumvents the privacy violations associated with video-based surveillance. In doing so, the system preserves the user’s dignity and offers enhanced acceptability in sensitive settings such as in elderly households. The ability to integrate intuitive monitoring and privacy protection into a non-contact radar-based system represents a significant and novel contribution and distinguishes this study from conventional methods.

## 3. Materials and Methods

### 3.1. Technical Architecture

This study aimed to develop a system capable of analyzing real-time changes in the position and posture of a subject in a non-contact manner to rapidly and accurately detect falls among the elderly. Existing fall detection systems based on wearable devices or camera surveillance each have limitations, such as discomfort resulting from the requirement to wear the device, limited battery life, and privacy concerns. With these issues in mind, we developed a non-contact fall detection system architecture centered on 4D imaging radar in this study.

The proposed system is based on the Retina-4sn radar instrument developed by Smart Radar System Inc. (SRS.Mobility LLC, Kissimmee, FL, USA),with its specifications presented in Table 1. The Retina-4sn is a high-resolution 4D imaging radar capable of providing four-dimensional information, including the position, velocity, size, and height of objects. These features make it particularly suitable for accurately analyzing various human postures such as standing, sitting, and lying down. The 4D imaging radar detects target movement and converts it into a set of points within a three-dimensional virtual space, thereby generating Point Cloud data. These data, which visually represent the location and shape of objects, enhance the efficiency of the analysis and support precise spatial interpretation.

Although the Retina-4sn radar includes a built-in AI module capable of distinguishing between standing, sitting, lying, and falling events, we did not utilize any of the internal classification functions of the radar in this study. Instead, we exclusively used the raw Point Cloud data acquired from the radar and developed an independent data processing pipeline, including customized preprocessing techniques (e.g., normalization, zero padding, and k-means clustering) and a novel CNN architecture tailored for posture classification. By following this custom-designed approach, we purposefully refrained from simply leveraging the existing capabilities of the radar, which highlights our technical contribution to this field of work.

To further improve the accuracy of distinguishing the action of lying down from an actual fall—an area where misclassification is common—fall detection was handled separately using a rule-based algorithm developed in this study. This algorithm was used to evaluate the changes in the position and velocity rather than relying on AI for classification.

In addition to movement and posture classification, to provide intuitive status monitoring, we implemented both a web-based monitoring interface and a Unity-based 3D visualization system. Even though conventional camera-based systems also enable non-contact fall detection, our radar-based approach maintains the advantage of intuitive monitoring without relying on video imaging, thereby avoiding privacy concerns. This distinguishes our system as a non-contact solution that protects the user’s privacy while maintaining real-time visualization and monitoring capabilities.

A systematic data collection environment for Point Cloud data was designed to develop an AI model for human posture classification. The radar was installed at a height of 2 m to optimize signal reception and minimize interference, with the experimental setup designed to cover a detection range of 7 m × 7 m. The azimuth and elevation angles were configured at 90° (±45°) to enable the contours of an object to be monitored across a wide range of angles both vertically and horizontally. With a rapid data update rate of 50 ms, the radar is capable of processing multiple frames per second, thereby ensuring real-time responsiveness in dynamic environments. The experimental environment was optimized to collect diverse posture data and considered the installation height, direction, and tilt for stable measurements.

As shown in Figure 1, the radar was installed at the height recommended by the manufacturer (2 m) to optimize the signal reception angle and minimize interference from obstacles. The radar was aligned frontally to precisely measure the target’s movement and reduce interference. Additionally, the tilt was set at −45° to ensure that the target was positioned at the center of the radar detection range. This facilitated reliable data acquisition for movements at ground level, such as when lying down. This configuration enabled the stable collection of data pertaining to various postural changes, thereby maximizing the efficiency and performance of the process of training the AI model. These settings align with the optimal tilt angle range (30–60°) reported in previous studies, further reinforcing the reliability of the experimental design [26].

### 3.2. Point Cloud Data Collection

In this study, Point Cloud data, which represent a three-dimensional virtual space for detecting and analyzing object movements, were collected within a range of 3 m on both sides and 7 m forward to capture diverse postures. To optimize the data quality, the environment was configured without obstacles to accurately capture target movements and minimize interference by external factors.

The experimental results showed that data stability decreased beyond a range of 2.5 m on either side. The detection distance was increased incrementally from 1 m to 7 m in intervals of 1 m, and the number and density of points in the Point Cloud data were quantified. The characteristics identified from the experimental results are presented in Table 2.

As is evident from the table, within the 1–3 m range, 700–1500 points were detected per frame, and this was sufficient to maintain uniform data distribution and high density. In particular, the 2–3 m range had the highest precision and stability, making it the optimal distance for capturing detailed movements such as posture classification. Unfortunately, at a distance of 1 m, despite the large number and high density of points, lower-body data were frequently lost. In the 4–5 m range, point intervals became irregular, and the density gradually decreased, whereas, beyond 6 m, the data quality sharply declined.

These experimental results clearly demonstrate that the distance between the target and the radar directly impacts the quality and reliability of the Point Cloud data. Therefore, setting an appropriate detection range is essential for collecting reliable data.

The data collected in this study provided a stable foundation for securing high-quality training datasets, which were used to maximize the performance of the AI model.

In this study, posture data were classified into the following three categories: standing, sitting, and lying down, as presented in Table 3, when using the Point Cloud for AI training. The Point Cloud data for the three categories were collected in the directions extending to the front, back, right, and left to minimize potential body occlusion effects. Irregular or unclear movements were included in the unknown category to maintain the data quality and improve the accuracy of the model.

The standing posture included the state of both standing still and the action of walking; thus, static and dynamic movement data were collected. The sitting posture was based on the state of sitting on a chair and included not only stable postures but also subtle movements. The lying-down posture was based on the state of lying down on a chair, with data collection including the action of lying down slowly. This can be used to distinguish between abrupt falls and the motion of lying down naturally.

The radar system processed data at 30 frames per second, and approximately 1500 frames were collected for each posture by capturing data for 50 s per posture. Data integrity was maintained by classifying ambiguous postures into the unknown category. This approach to dataset construction was designed to precisely classify posture data by minimizing false positives and missed detections. Data were captured in various environments and under various conditions to ensure that the AI model maintains high reliability and accuracy in real-world applications. This is expected to enhance the effectiveness of radar-based posture analysis systems and contribute to their applicability in practical use cases.

### 3.3. Point Cloud Data Preprocessing

The Point Cloud data received from the radar sensors were systematically preprocessed to ensure that the AI model was successfully trained. Preprocessing is required because Point Cloud data contain high-dimensional and complex information, including information about the position, velocity, time, and Doppler power. Using these raw data as direct input into AI models can lead to issues such as overfitting or reduced learning efficiency. Therefore, preprocessing is necessary to remove noise, extract the key information required by the learning model, and optimize the performance and stability.

The preprocessing method developed in this study utilized only the spatial position information directly related to posture classification within the Point Cloud data, specifically the x-, y-, and z-coordinate data. This approach removed unnecessary information to enhance the data processing speed, ensure real-time performance, and improve the processing efficiency of the model. Simplifying the data lowers the computational cost and model complexity while maximizing the accuracy and practicality.

The number of samples generated in the form of Point Cloud data varies depending on the target’s actions and movements. For example, rapidly moving targets generate more points, whereas static targets result in fewer points. This data imbalance could render the input data less consistent and would negatively affect the learning performance. To address this issue, normalization and balance adjustment processes were implemented to improve the quality of the training data to enable the model to operate more reliably across diverse behavioral patterns.

The data were collected over 50 s, during which 1500 frames were captured, with the number of points per frame (*N*) ranging from 200 to 1500 (Table 2). Considering that the data used as input for deep learning models must maintain a consistent size, frames with varying lengths were normalized to *N* = 500. Frames with fewer than 500 points (*N* ≤ 500) were padded with zeros using the back-padding technique by filling the missing portions with the value “0” to avoid introducing artificial values. The point (0, 0, 0) in the three-dimensional space has clearly distinguishable spatial characteristics and does not have a significantly negative impact on model training; thus, other than zero padding, further post-processing was not performed.

The transformation of Point Cloud data with varying lengths ensured that the input data were consistent to improve the learning efficiency such that the data were suitable for deep learning, particularly to enhance the stability and performance during training and inference.

For frames with *N* > 500, the k-means clustering algorithm was applied to reduce redundancy while preserving the critical spatial characteristics. Specifically, the Point Cloud data were divided into k = 500 clusters, and a new frame was constructed by extracting the centroid of each cluster, which resulted in uniformly sized input. This process minimizes unnecessary data while retaining the original structure of the key spatial features. The clustering operation aims to minimize the sum of the squared distances between each point and its assigned cluster centroid, as expressed in Equation (1):(1)$$\arg\underset{C}{\min}\sum_{j=1}^{k}\sum_{p_{i}\in C_{j}}{\left\|p_{i}-\mu_{j}\right\|}^{2}$$
where $p_{i}$ represents a data point in the point cloud, $\mu_{j}$ is the centroid of cluster $C_{j}$, and k = 500 is the predefined number of clusters. This optimization ensures that points within the same cluster are as close as possible to their centroid, thereby enabling effective data compression while maintaining geometric fidelity [27].

As shown in Figure 2, before the application of k-means clustering, the original Point Cloud data were irregularly distributed, and this could adversely affect the learning efficiency of AI models. However, as demonstrated in Figure 2, the Point Cloud data (*N* = 500) generated after clustering effectively preserved the primary spatial features of the original data and reduced irregularity. This improvement was expected to further enhance the stability and performance of the learning model. Radar signals may distort the data sequence during the collection process due to factors such as the time required for the signal to complete a round-trip, movement of the target, and changes in the distance to the radar equipment. Particularly, data sequences collected from subtle body movements or rotations may appear random and could potentially degrade the performance of models designed to learn the location pattern, such as CNNs. This issue was addressed by applying a systematic data sorting method. The reconstruction of data sequences to ensure consistency enhances the ability of the learning model to efficiently learn patterns. Ultimately, this improves the model training performance and prediction accuracy. The Point Cloud data in each frame are expressed as coordinate values in 3D space, (*x_i_*, *y_i_*, *z_i_*), represented as *p_i_* = (*x_i_*, *y_i_*, *z_i_*), *i* = 1, 2, …, *N*. To systematically sort these data, a method that sorts the data in ascending order using Equation (2) was applied to the Point Cloud data *p_i_* [28].(2)$$\left(x_{a},y_{a},z_{a}\right)≺\left(x_{b},y_{b},z_{b}\right)\;\Leftrightarrow \;\left\{\begin{array}{l} x_{a}<x_{b}\\ or\;(x_{a}=x_{b}\wedge y_{a}<y_{b})\\ or\;(x_{a}=x_{b}\wedge y_{a}=y_{b}\wedge z_{a}<z_{b}) \end{array}\right.$$

Equation (2) defines the lexicographical order between the following two vectors: (*x_a_*, *y_a_*, *z_a_*) and (*x_b_*, *y_b_*, *z_b_*). This order establishes a rule to determine whether one vector is “smaller” or “greater” than the other. If the first coordinate *x_a_* < *x_b_*, then (*x_a_*, *y_a_*, *z_a_*) is considered “smaller”. If *x_a_* = *x_b_*, the data are sorted based on *y_a_* < *y_b_*. Lastly, if *x_a_* = *x_b_* and *y_a_* = *y_b_*, the data are sorted based on *z_a_* < *z_b_*. The sorting criteria involve first sorting all points in the ascending order of their *x*-coordinates. For points with identical *x*-coordinates, they are further sorted in the order in which their *y*-coordinates ascend. Finally, for points with identical *x*- and *y*-coordinates, sorting is conducted in the ascending order of their *z*-coordinates.

This sorting process is illustrated through the pre- and post-sorted states in Table 4. During this process, the actual distances or relative positions between the points remain unchanged, and only the order in which the data are input is rearranged. This sorting method systematically organizes the sequence of Point Cloud data to ensure consistency in the input order for the CNN model. This allows the model to effectively learn patterns and maximize its performance.

### 3.4. Design of the Artificial Intelligence Model

In this study, the CNN architecture (illustrated in Figure 3) was designed to classify three postures (standing, sitting, or lying) for fall detection. This CNN architecture effectively learns the complex spatial features of Point Cloud data to achieve high classification accuracy [14].

In the preprocessing stage, techniques such as zero padding and k-means clustering were used to normalize and balance the Point Cloud data to lower the amount of noise and establish a stable learning environment. This process transforms the input data into a form suitable for CNN training.

In the feature extraction stage, the Point Cloud data are mapped into a 2D array, followed by two stages of 2D convolution layers for progressive feature learning. The first convolution layer uses 16 channels to capture basic positional and spatial features, while the second layer uses 64 channels to learn more abstract and complex features. Each convolution layer employs the Leaky ReLU activation function to handle negative values effectively, and pooling layers reduce the amount of data while retaining essential information to enhance the computational efficiency.

In the classification stage, the extracted features are transformed into a 1D vector through a flatten layer, and posture classification is performed using a fully connected layer. During this process, dropout (0.5) was applied to randomly deactivate some neurons during training, thereby serving to lower the dependency on specific neurons and prevent overfitting. Finally, the output layer, with a Softmax activation function, provides the predicted probabilities for the following three classes: standing, sitting, or lying.

The CNN model was trained using the Adam optimization algorithm with the initial learning rate set to 0.001. The batch size was set to 32, and training was conducted over a total of 100 epochs. Categorical cross-entropy was used as the loss function to address the multi-class classification problem. During the training process, both the training loss and validation loss were monitored to assess model convergence and evaluate the possibility of overfitting.

The data sorting and normalization techniques that were applied in this study eliminated randomness in the Point Cloud data and maintained a consistent input structure. This contributed to securing the stability needed for the learning model to effectively learn patterns to maximize its performance.

### 3.5. Comparative Analysis of Fall Detection

To effectively detect falls, the establishment of clear criteria that distinguish between the normal movement of lying down slowly and falling is essential. During a fall, the height (z_max_) of the body part farthest from the ground (the crown of the head) and the velocity change at this height are prominent features for identification. Based thereupon, we developed a method for analyzing and comparing the z_max_ values and velocity changes of z_max_ to differentiate between the actions of lying down slowly and falling.

Figure 4 illustrates that during movement that involves lying down slowly (standing → lowering posture (A) → sitting (B) → leaning the upper body forward (C) → lying down), the z_max_ value gradually decreases over time. In the “standing → lowering posture → sitting” segment (B), z_max_ decreases consistently, whereas, in the “sitting → leaning the upper body forward” segment (C), an intentional posture adjustment to maintain body stability causes z_max_ to momentarily increase before decreasing again. This pattern indicates that the body maintains its balance in a controlled manner during the process of lying down slowly.

In contrast, Figure 4 shows that during a fall, z_max_ decreases abruptly without intermediate stages, and this movement is characterized by significant fluctuations. Because falls occur in an uncontrolled manner, the body rapidly approaches the ground within a short time, resulting in this characteristic pattern. The sudden, sharp changes in z_max_ during a fall distinctly differ from the gradual changes observed for movement when lying down normally. These differences make the z_max_ value an effective indicator for distinguishing between normal movements and falls.

Additionally, Figure 5 compares the velocity changes at z_max_ between the movements of lying down slowly and falling. In Figure 5, during the process of lying down slowly, the velocity of z_max_ exhibits a relatively gradual and consistent pattern over several hundred frames. In contrast, during falling, the velocity of z_max_ changes rapidly and irregularly within a very short period of time. This velocity pattern reflects the involuntary and uncontrolled nature of falls and serves as a critical criterion for distinguishing between movements associated with lying down slowly and falling.

This study quantitatively analyzed the changes in z_max_ and velocity at z_max_ to propose clear criteria for differentiating between movement when lying down slowly and movement when falling. This analysis contributes to improving the accuracy of real-time fall detection systems and provides a foundation for efficient application in various real-life scenarios.

### 3.6. System Configuration for Fall Detection Monitoring

The fall detection monitoring system implemented in this study consists of the following components. First, as shown in Figure 6a, the Point Cloud coordinate data of the target detected by the radar are used as input for the AI model. The AI model analyzes the input data and outputs the target’s state and their x- and y-coordinates, which are transmitted to a specific port via user datagram protocol (UDP) socket communication. As shown in Figure 6b, the output data are captured in packet form through a Python program (Version 3.11), processed (logged), and stored in a database via an API. Simultaneously, the data are also transmitted to the Unity-based monitoring program via UDP socket communication. If a fall event is captured, a real-time event is triggered on both the web page and the monitoring program through Server-Sent Event (SSE) communication to generate immediate alerts. The overall architecture of the platform, presented in Figure 6c, consists of the Unity-based monitoring program and an administrator web page. The Unity program moves a 3D avatar within virtual space mapped to the real-world coordinates based on the received state and the x- and y-coordinate information and applies animations according to the detected posture changes to provide real-time monitoring. Additionally, the React-based administrator web page enables users to manage observation targets and check their log information in real time. When a fall event is registered, the immediate alerts that are generated to both the Unity program and the web page enable prompt responses.

## 4. Principal Result

### 4.1. Performance Analysis and Evaluation of the Posture Classification Model

The performance of the CNN model that was used for classifying the three postures (standing, sitting, or lying) was optimized for fall detection using 5-fold cross-validation. Figure 7 visually illustrates the 5-fold cross-validation process and the performance evaluation results for each iteration.

The 5-fold cross-validation process involves dividing the data into five folds, using one fold (black) as the test set and the remaining four folds (gray) as the training set during each iterative cycle. The test fold is rotated cyclically across iterations to ensure that each fold is used once as test data. In each fold $i\;\left(i=1,\;2,\;\ldots ,\;5\right)$, the test accuracy ${\mathrm{Accuracy}}_{i}$ was calculated using Equation (3) as follows:(3)$${\mathrm{Accuracy}}_{i}=\frac{{Number\;of\;correctly\;classified\;samples\;in\;fold}_{i}}{{Total\;number\;of\;test\;samples\;in\;fold}_{i}}$$

After completing all five folds, the final test accuracy was computed by averaging the accuracy values from each fold according to Equation (4):(4)$$Final\;Accuracy=\frac{1}{5}\sum_{i=1}^{5}{Accuracy}_{i}\;$$

This approach provides a comprehensive evaluation of the generalization ability of the model by ensuring that every sample is used once for testing.

As shown in Figure 7, the test fold was rotated over five iterations, and the test accuracy of the CNN model was evaluated iteratively in each cycle. The test accuracy consistently exceeded 98% across all iterations, indicating the strong predictive performance of the model even with varying data distributions. The average test accuracy, calculated to be 98.66%, demonstrated the exceptional performance and robust generalization capability of the CNN model in fall detection and posture classification tasks. These results confirmed that the proposed CNN model delivers reliable performance in fall detection and posture classification.

### 4.2. Experimental Validation and Accuracy Analysis of the Fall Detection Algorithm

The accuracy of the proposed fall detection algorithm was evaluated by conducting experiments by hanging a black curtain in front of the radar system (Figure 8). Participants randomly performed motions that involved lying down slowly or falling, during which the changes in z_max_ and the velocity of these changes were measured.

A total of 10 volunteers participated in the experiments, with each of them randomly performing movements that involved lying down slowly or falling. The measured frame counts revealed that motions associated with lying down primarily required 80–90 frames to capture in their entirety, whereas fall motions occurred over approximately 20–30 frames.

Analysis of the detection accuracy presented in Table 5 revealed that the accuracy for detecting “lying down motion” was 100%, whereas that for the “fall motion” was 95%. Overall, out of a total of 20 movements (10 lying down + 10 falls) performed by the participants, 19 cases were detected correctly, resulting in an overall accuracy of 95%. Furthermore, as shown in Table 6, velocity change analysis showed that the motions undertaken when lying down slowly involved gradual changes within a range of ±0.8 m/s, whereas the motion of falling entailed rapid changes at ±2.3 m/s.

In conclusion, the motion of lying down slowly involved gradual velocity changes over a longer period (80–90 frames), whereas falling occurred by way of rapid and significant velocity changes for a shorter duration (20–30 frames). This clear distinction demonstrates the ability of the algorithm to differentiate between the two motions. The results highlight the practical applicability of the fall detection system in various user environments, suggesting that radar technology is a promising alternative or complement to conventional vision-based systems.

### 4.3. Comparative Evaluation of Posture Classification and Fall Detection Performance

Table 7 and Table 8 present a comparative summary of recent radar-based studies on human posture classification and fall detection, respectively. These studies vary in terms of the number of classified states, model types, and application contexts. However, they share the common objective of achieving reliable performance in non-contact sensing environments.

As indicated in Table 7, the posture classification accuracy ranges from 84.94% to 99.25% across different approaches. The Bi-LSTM and MLP-based models performed strongly in both dynamic and static scenarios. In our study, the proposed method using the CNN architecture optimized for preprocessed Point Cloud data classified the three postures (standing, sitting, or lying) with an accuracy of 98.66%, which is comparable to or higher than that of most existing methods. Unlike some prior studies that relied on complex temporal models or handcrafted features, our approach maintains computational efficiency and is well suited for real-time use on embedded systems.

Similarly, the previous fall detection studies in Table 8 report accuracies ranging from 87.47% to 98.83%. Although most of these studies employed deep learning models such as Bi-LSTM, XGBoost, or VRAE, our study adopts a lightweight rule-based algorithm that uses the changes in z_max_ and the velocity. Our method, which achieved 95% fall detection accuracy, demonstrates competitive performance with significantly lower computational complexity. In contrast to black-box AI models, our algorithm is interpretable, and the fact that it does not require training makes it adaptable for deployment in diverse environments.

Taken together, the results confirm that the proposed system delivers posture classification and fall detection performance on par with or superior to that of existing state-of-the-art systems while offering the advantages of model simplicity, privacy protection, and real-time applicability. Moreover, the integration of a Unity-based avatar visualization interface further differentiates our system by enabling intuitive, video-free monitoring—a feature rarely addressed in previous studies.

### 4.4. Fall Detection Monitoring System

Based on the design of the AI model, we developed a monitoring system that combines the three-posture (standing, sitting, or lying) classification model with a fall detection algorithm that accepts radar Point Cloud data for processing. The system—built on a web page and the Unity engine development platform—enables the subject’s condition to be monitored in real time and provides immediate alerts to the administrators in the event of a fall.

The web-based fall detection system allows users to register and view subject information, monitor the subject’s location and condition in real time, and send immediate notifications to registered emergency contacts upon detecting a fall. As shown in Figure 9 and Figure 10, the subject’s location and status are visualized as a heatmap based on the duration of stay and a set of pictograms to enable their movement paths to be intuitively tracked. This functionality is valuable for predicting health anomalies or emergency situations in advance.

The avatar-based user interface, developed using the Unity game engine, is designed to provide an intuitive real-time view of the subject’s location and condition. As depicted in Figure 11, the avatar reflects the subject’s actual movements while respecting their privacy. This implementation eliminates the need for additional physical devices by adding virtual cameras within the Unity Engine to enable the subject’s status to be monitored from various angles and effectively eliminating blind spots.

This system provides an environment that allows the user to accurately assess the subject’s condition and take prompt action when necessary. By sending immediate alerts during a fall, the system enables rapid responses to emergencies while protecting an individual’s privacy by using radar data instead of video footage. Additionally, the extensibility of the Unity game engine allows for flexible design tailored to diverse user environments and requirements.

In conclusion, the proposed system overcomes the limitations of traditional fall detection technologies by offering a practical and efficient solution for real-time monitoring and emergency response.

## 5. Discussion

In this study, a real-time monitoring system for human posture classification and fall detection was developed and validated by integrating non-contact radar sensors with AI technology. The proposed CNN model achieved an average posture classification accuracy of 98.66%, while the fall detection algorithm attained an accuracy of 95%, as shown in Figure 7 and Table 5. Through the analysis of z_max_ and the velocity changes illustrated in Figure 4 and Figure 5, the system was able to effectively distinguish between the movements of lying down slowly and actually falling. These results demonstrate that the system successfully addressed the initial research objectives, namely, alleviating user discomfort, protecting privacy, and enabling real-time monitoring.

Compared to existing wearable device-based or vision-based systems, this study advanced human monitoring technology by presenting a real-time monitoring solution that ensures non-contact operation, privacy protection, and reliable performance in non-line-of-sight (NLOS) environments. In particular, the implementation of a 3D avatar-based interface without video recording, as visualized in Figure 11, represents a significant improvement over conventional approaches by realizing both intuitive monitoring and enhanced personal privacy protection.

However, as this study was conducted in a controlled experimental environment, further validation in diverse home settings is necessary. In particular, additional experiments would have to be conducted in environments that present real-world challenges, such as those with metallic structures and electromagnetic interference, to enhance the reliability of the system. Additionally, research is needed to address the potential occurrence of false alarms caused by sudden movements that are not actual falls.

Future research should focus on collecting additional data that reflects home environments and user conditions, as well as expanding the training dataset to improve the generalization performance of the AI model. Furthermore, analyzing the user’s movements after a fall and introducing a stepwise classification approach, such as “suspected fall” and “confirmed fall”, could effectively reduce false alarms. Additionally, further research is necessary to enhance the capability of the system to monitor and analyze users’ behavior and changes in their physical condition in real time over extended periods. In particular, incorporating the concept of temporal variability—such as gradual behavioral drift or physiological changes over time—can improve the adaptability and long-term reliability of the system for health monitoring applications [29]. Through such advancements, the system could evolve beyond short-term fall detection into a comprehensive health management platform capable of tracking changes in the long-term health status and providing early warnings. Such advancements would transform the system from merely detecting falls to a comprehensive safety management solution that monitors the overall health status and provides early warnings for life-threatening situations.

This system has significant potential to address societal challenges, such as the increase in single-person households in aging societies, and could serve as a critical technological foundation for such solutions. By demonstrating the technical feasibility of a real-time monitoring system, this study provides valuable foundational data for commercialization by incorporating advanced technology and diverse real-world scenarios.

## 6. Conclusions

In this study, we developed and experimentally validated a real-time monitoring system that integrates 4D imaging radar sensors and AI-based analysis techniques to classify human postures and detect falls in a non-contact manner. The proposed CNN-based posture classification model achieved a high average accuracy of 98.66%, and the fall detection algorithm reached an accuracy of 95% to successfully address the primary objectives of this study as follows: reducing user discomfort, protecting privacy, and enabling real-time monitoring.

Although conventional camera-based fall detection systems provide intuitive monitoring, they are often criticized for infringing on user privacy because they are designed to capture video material. In contrast, this study implemented a Unity-based avatar visualization system, which enables intuitive monitoring without video, thereby maintaining the advantages of existing systems while effectively mitigating their limitations. This approach offers a meaningful technological advancement, particularly for older adults or individuals for whom privacy is a critical concern.

From a technical standpoint, this study proposed a lightweight architecture that utilizes normalized radar-derived Point Cloud data, processed through k-means clustering and zero padding, and classified via a CNN model. In addition, we implemented a rule-based fall detection algorithm that uses the z_max_ values and velocity changes to detect falls. Notably, the ability to distinguish between the movements associated with lying down slowly and actually falling—often misclassified in prior studies—represents a meaningful achievement.

The system demonstrated stable motion detection performance through everyday obstacles such as clothing, furniture, and thin walls, and also maintained high reliability in non-line-of-sight (NLOS) environments. However, in real-world residential settings, electromagnetic interference from appliances and electronic devices may affect the quality of radar signals. This issue was not addressed in the current study and should be explored in future research.

This study showed that human posture can be accurately classified using radar-based Point Cloud data without complex skeletal estimation. Moreover, it demonstrated that intuitive and user-friendly monitoring is possible even without using video. By integrating a web-based interface and Unity-based visualization, the system not only enables real-time monitoring but also lays the groundwork for extended functions such as data-driven health tracking and analysis.

Importantly, the Unity-based avatar system holds potential beyond basic visualization. It could be expanded into a metaverse or digital twin–based virtual monitoring platform in the future. The ability to reflect users’ movements and status in real time within a virtual space suggests that this system could evolve from a simple fall detection tool into a long-term health monitoring and personalized care platform.

In conclusion, this paper presents an integrated solution that balances technical robustness and real-world applicability in the fields of human status recognition and fall detection. It offers valuable foundational technology for applications in elderly care, digital healthcare, and real-time smart home safety systems.

## Figures and Tables

**Figure 1 sensors-25-03452-f001:**
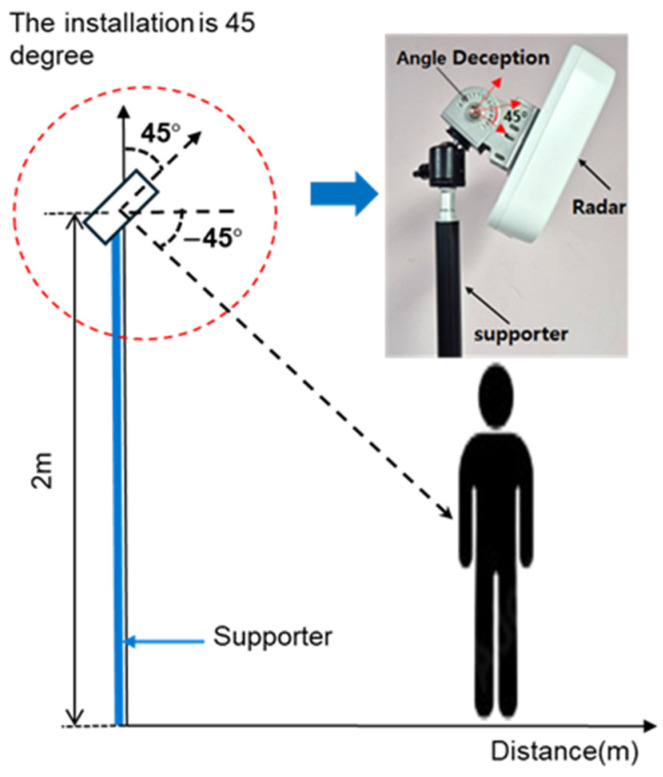
Radar system installation diagram for experiments.

**Figure 2 sensors-25-03452-f002:**
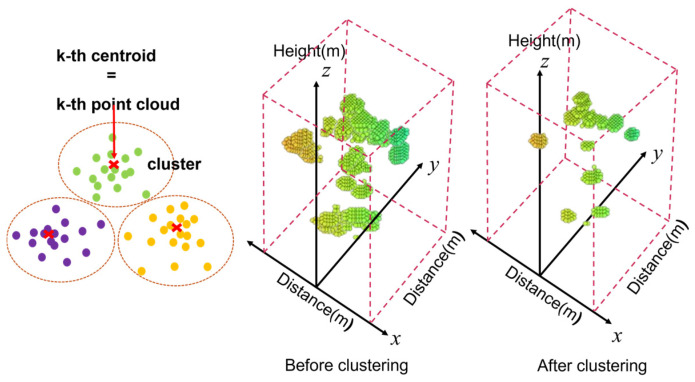
Visualization of clustering results in 3D Point Cloud space: overview of K-means clustering and comparative analysis of data distribution before and after clustering.

**Figure 3 sensors-25-03452-f003:**
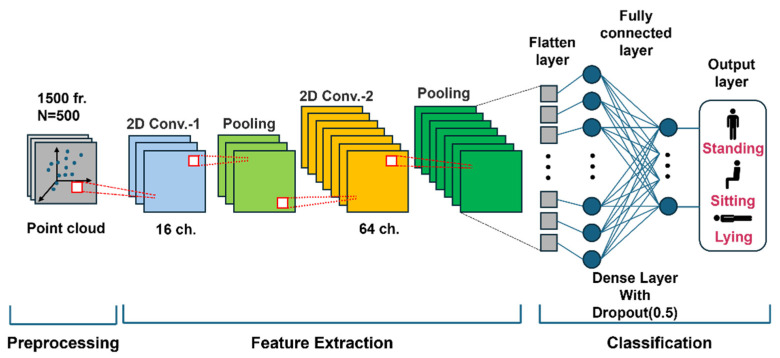
Schematic representation of the architecture of the CNN model designed for classifying three postures (standing, sitting, or lying).

**Figure 4 sensors-25-03452-f004:**
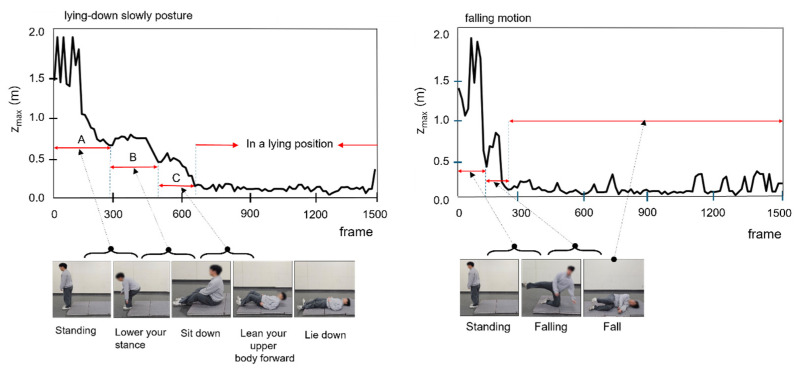
Analysis of the changes in z_max_ over time: comparison of postural transitions when lying down slowly and falling.

**Figure 5 sensors-25-03452-f005:**
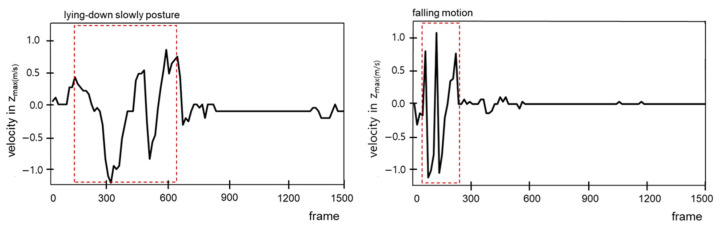
Analysis of velocity changes at z_max_: comparison of lying down slowly and falling.

**Figure 6 sensors-25-03452-f006:**
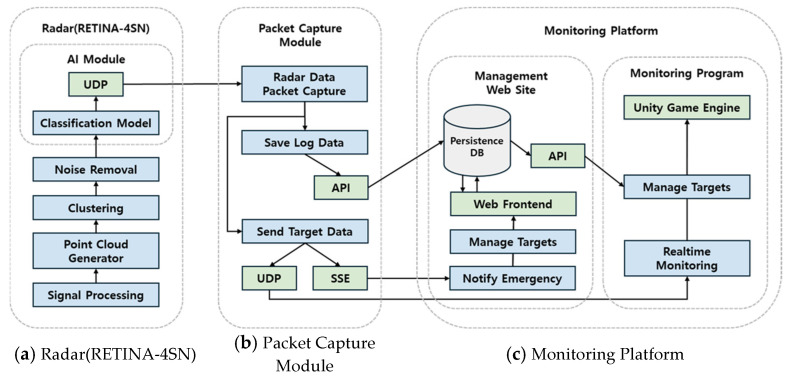
Diagrammatic representation of the monitoring system configuration.

**Figure 7 sensors-25-03452-f007:**
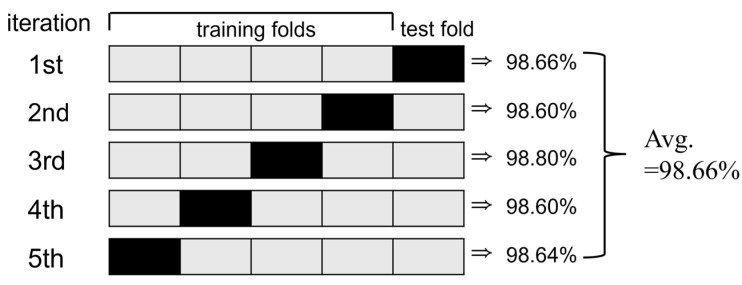
K-fold cross-validation result for training and testing.

**Figure 8 sensors-25-03452-f008:**
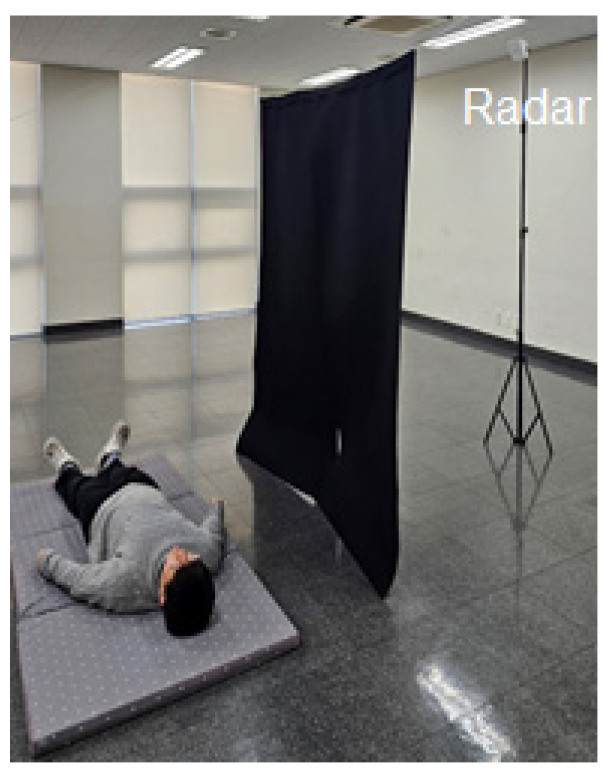
Fall detection experiment in a blind environment.

**Figure 9 sensors-25-03452-f009:**
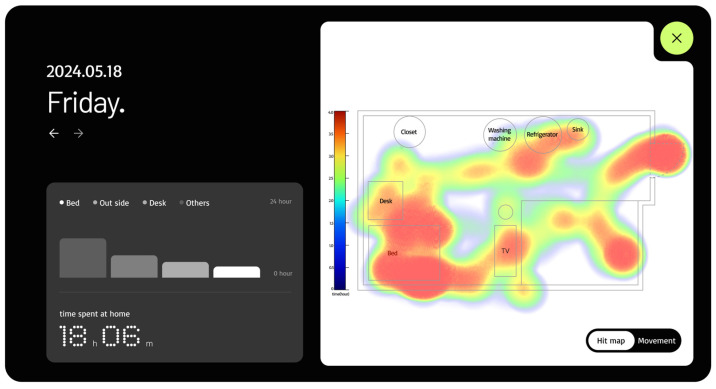
Heatmap showing the position and status of the subject on the web page.

**Figure 10 sensors-25-03452-f010:**
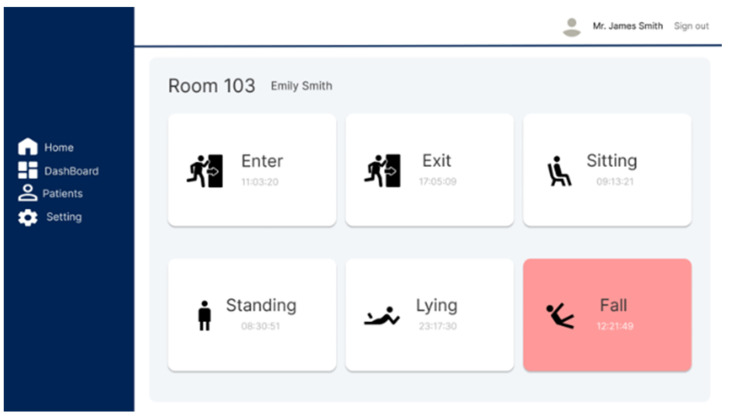
Pictogram showing the position and status of the subject on the web page.

**Figure 11 sensors-25-03452-f011:**
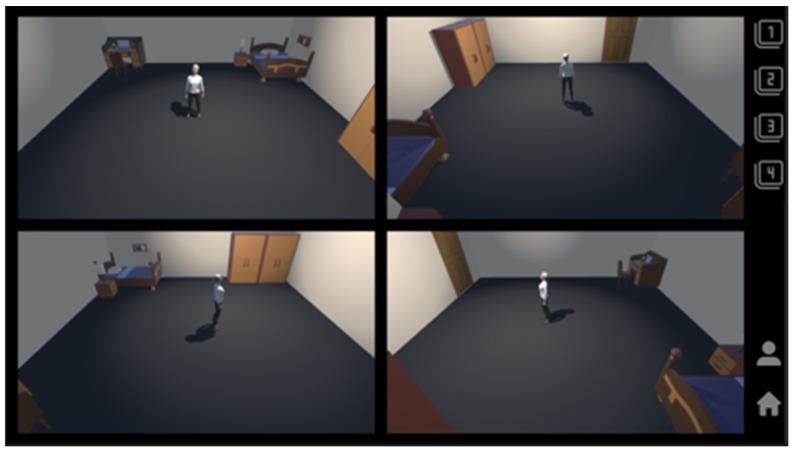
Real-time position and status using avatars in the Unity Engine.

**Table 1 sensors-25-03452-t001:** Performance of 4D imaging radar (Retina-4ns).

Item	Description	Picture
Frequency	77~81 GHz	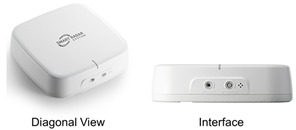
Detection range/area	12 m/7 m × 7 m	
Azimuth angle FOV	90° (±45°)	
Elevation angle FOV	90° (±45°)	
Update rate	50 ms	
Communication interface	Wi-Fi	

**Table 2 sensors-25-03452-t002:** Point Cloud count and density by distance from radar.

Distance from Radar	No. of Points in the Point Cloud	Quality of Point Cloud
1 m	900~1500	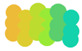
2 m	800~1400	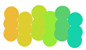
3 m	700~1400	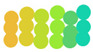
4 m	700~1300	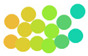
5 m	600~1200	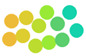
6 m	400~1100	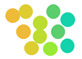
7 m	200~900	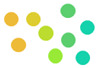

**Table 3 sensors-25-03452-t003:** Visualization of Point Cloud data by posture.

	Actual Photograph	Point Cloud
Standing		
	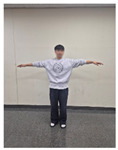	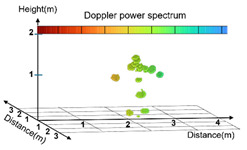
Sitting		
	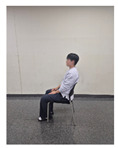	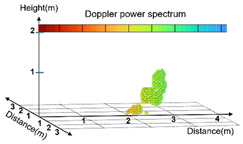
Lying		
	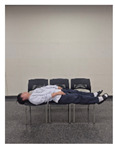	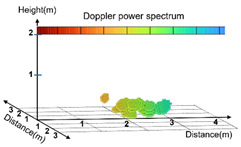

**Table 4 sensors-25-03452-t004:** Comparison of Point Cloud data before and after alignment of the x-, y-, and z-coordinates.

	Frame	*x*	*y*	*z*
Before sorting				
	716	1.24	1.99	1.57
	716	1.96	1.99	1.42
	716	0.48	0.11	0.96
	716	1.61	0.17	0.69
	716	1.50	0.58	1.40
	716	1.96	1.99	0.67
	716	0.26	0.48	0.12
	716	0.36	0.48	1.82
	716	0.36	0.48	1.97
	716	0.90	0.44	0.55
	716	1.61	1.60	0.67
	716	0.90	1.21	1.80
	716	1.61	1.60	0.55
	716	1.96	1.99	1.99
After sorting				
	716	0.26	0.48	0.12
	716	0.36	0.48	1.82
	716	0.36	0.48	1.97
	716	0.48	0.11	0.96
	716	0.90	0.44	0.55
	716	0.90	1.21	1.80
	716	1.24	1.99	1.57
	716	1.50	0.58	1.40
	716	1.61	0.17	0.96
	716	1.61	1.60	0.55
	716	1.61	1.60	0.67
	716	1.96	1.99	0.58
	716	1.96	1.99	0.67
	716	1.96	1.99	1.42

**Table 5 sensors-25-03452-t005:** Detection accuracy analysis for lying down slowly or falling based on the frame count and changes in the position of z_max_.

	Volunteer	z_max_ (m)	Frame	State	Detection
Lying down slowly					
	1	0~1.8	90	lying	lying
	2	0~1.7	90	lying	lying
	3	0~2.0	90	lying	lying
	4	0~1.8	90	lying	lying
	5	0~1.7	80	lying	lying
	6	0~1.8	80	lying	lying
	7	0~1.7	90	lying	lying
	8	0~1.9	80	lying	lying
	9	0~2.0	80	lying	lying
	10	0~1.8	80	lying	lying
Falling					
	1	0~1.7	20	fall	fall
	2	0~1.8	20	fall	fall
	3	0~2.0	30	fall	fall
	4	0~1.9	20	fall	fall
	5	0~1.7	30	fall	lying
	6	0~1.8	20	fall	fall
	7	0~2.0	20	fall	fall
	8	0~1.9	30	fall	fall
	9	0~1.7	30	fall	fall
	10	0~1.7	25	fall	fall

**Table 6 sensors-25-03452-t006:** Detection accuracy analysis for lying down slowly or falling based on the frame count and changes in the velocity of z_max_.

	Volunteer	Velocity (m/s)	Frame	State	Detection
Lying down slowly					
	1	−0.51~+0.65	90	lying	lying
	2	−0.43~+0.65	90	lying	lying
	3	−0.58~+0.72	90	lying	lying
	4	−0.72~+0.80	90	lying	lying
	5	−0.58~+0.80	80	lying	lying
	6	−0.51~+0.80	80	lying	lying
	7	−0.72~+0.80	90	lying	lying
	8	−0.65~+0.72	80	lying	lying
	9	−0.72~+0.72	80	lying	lying
	10	−0.80~+0.65	80	lying	lying
Falling					
	1	−2.32~+2.10	20	fall	fall
	2	−1.09~+1.88	20	fall	fall
	3	−2.17~+2.25	30	fall	fall
	4	−1.09~+2.25	20	fall	fall
	5	−0.72~+1.88	30	fall	lying
	6	−0.58~+2.17	20	fall	fall
	7	−2.32~+1.74	20	fall	fall
	8	−1.01~+1.74	30	fall	fall
	9	−0.87~+2.25	30	fall	fall
	10	−0.58~+1.59	25	fall	fall

**Table 7 sensors-25-03452-t007:** Comparison of existing posture classification studies based on number of classes, classification models, and accuracy.

Study	Classified Postures (Classes)	AI Model	Reported Accuracy
Baird et al. (2018) [15]	3 (standing, sitting, and lying)	Decision Tree	84.94%
Shrestha et al. (2020) [16]	6 (walking, sitting, standing, picking, drinking, and falling)	Bi-LSTM	91%
Liang et al. (2021) [17]	4 (empty, walking, standing, and falling)	Bi-LSTM	99.25%
Zhang et al. (2023) [18]	6 (hands up, horse stance, lunge, lying, sitting, and standing)	MLP	94%
Werghi et al. (2002) [19]	19 postures	Bayesian + Wavelet Transform	98%
The proposed study	3 (standing, sitting, and lying)	CNN	98.66%

**Table 8 sensors-25-03452-t008:** Comparison of radar-based fall detection studies based on model type and reported accuracy.

Study	Detection Method/Model	Reported Accuracy
Zhang et al. (2024) [21]	CNN + Bi-LSTM	98.83%
Liu et al. (2023) [22]	4-stage Hierarchical Algorithm (CW Radar)	93.24%
Wu et al. (2023) [23]	SPWVD Feature + XGBoost	87.47%
Gong et al. (2023) [24]	DBSCAN + State Transition Prediction	96.3%
Xie et al. (2020) [25]	VRAE (Variational RNN AutoEncoder)	98%
Proposed Study	Rule-based (zmax and velocity change)	95%

## Data Availability

Dataset available on request from the authors.

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
