# Peer review of "Non-Contact Fall Detection System Using 4D Imaging Radar for Elderly Safety Based on a CNN Model"

_sensors, 2025, doi:10.3390/s25113452_

Round 1
Reviewer 1 Report
Comments and Suggestions for Authors
The author proposes a non-contact fall detection system using a 4D imaging radar sensor with AI (CNN). The reviewer has some comments on the manuscript.
- The novelty and contribution are not clearly displayed.
- The author should conduct a comprehensive review of the fall detection system and other radar-based detection systems. For example, (1) radar-based detection system - Cai, Fulin, et al. "STRIDE: Systematic radar intelligence analysis for ADRD risk evaluation with gait signature simulation and deep learning." IEEE sensors journal 23.10 (2023): 10998-11006. (2) AI method for radar sensing - Cai, Fulin, Teresa Wu, and Fleming YM Lure. "E-BDL: enhanced band-dependent learning framework for augmented radar sensing." Sensors 24.14 (2024): 4620. Since the contribution and novelty are unclear, the reviewer cannot give further suggestions. Novelty on the radar equipment or the entire system.
- The data preprocessing for the CNN model is unclear. How is the 4D data transformed into 2D for learning? The rationale for the design and also why not using the existing pre-trained model.
- For the experiment, why is there an evaluation of posture, but the title of the paper is about fall detection?
- Is the experiment subject-independent or dependent according to the data split?
- The simulation method of fall detection should be briefly introduced.
- The Discussion and Conclusion must be rewritten to discuss and show insight into the aim of this study.
- For future work, the author should consider long-term and real-time monitoring, which shows temporal variation [1] [1] Cai, Fulin, et al. "TP-CL: A novel temporal proximity contrastive learning approach for obstructive sleep apnea detection using single-lead electrocardiograms." Biomedical Signal Processing and Control 100 (2025): 106993.
Author Response
Comments 1: The novelty and contribution are not clearly displayed.
Response 1: In response to the reviewer’s comments, we have revised the Abstract and Introduction to clearly highlight the originality and contributions of this study.
Comments 2: The author should conduct a comprehensive review of the fall detection system and other radar-based detection systems. For example, (1) radar-based detection system - Cai, Fulin, et al. "STRIDE: Systematic radar intelligence analysis for ADRD risk evaluation with gait signature simulation and deep learning." IEEE sensors journal 23.10 (2023): 10998-11006. (2) AI method for radar sensing - Cai, Fulin, Teresa Wu, and Fleming YM Lure. "E-BDL: enhanced band-dependent learning framework for augmented radar sensing." Sensors 24.14 (2024): 4620. Since the contribution and novelty are unclear, the reviewer cannot give further suggestions. Novelty on the radar equipment or the entire system.
Response 2: In response to the reviewer’s comments, we have expanded the Related Work section and clearly described the originality and contributions of this study.
Comments 3: The data preprocessing for the CNN model is unclear. How is the 4D data transformed into 2D for learning? The rationale for the design and also why not using the existing pre-trained model.
Response 3: To enhance the clarity of the research methodology, we have added a detailed explanation in Section 3.3 (Point Cloud Data Preprocessing) regarding the rationale and process of converting the Point Cloud data into 2D.
Comments 4: For the experiment, why is there an evaluation of posture, but the title of the paper is about fall detection?
Response 4: This method was applied to distinguish between daily postures such as standing, sitting, and lying, and fall events.
Comments 5: Is the experiment subject-independent or dependent according to the data split?
Response 5: To maintain independence between subjects, the data were split on a per-subject basis during the 5-fold cross-validation process.
Comments 6: The simulation method of fall detection should be briefly introduced.
Response 6: In Section 4.2 (Performance Analysis and Evaluation of the Posture Classification Model), we described the simulation method used for fall detection.
Comments 7: The Discussion and Conclusion must be rewritten to discuss and show insight into the aim of this study.
Response 7: In response to the reviewer’s feedback, we revised the Discussion and Conclusion sections to improve the clarity of result interpretation and enhance the overall quality of the manuscript.
Comments 8: For future work, the author should consider long-term and real-time monitoring, which shows temporal variation [1] [1] Cai, Fulin, et al. "TP-CL: A novel temporal proximity contrastive learning approach for obstructive sleep apnea detection using single-lead electrocardiograms." Biomedical Signal Processing and Control 100 (2025): 106993.
Response 8: In the Conclusion section, we suggested long-term monitoring that reflects changes over time as a future research direction.

Reviewer 2 Report
Comments and Suggestions for Authors
Please refer to the attachment

Author Response
Comments 1: The abstract can be improved. The authors need to clarify that the challenge addressed in this manuscript is how 4D radar sensors can accurately identify falls in the elderly, rather than existing wearable devices. In addition, authors should provide key technical details used to address the challenges, including unique system structures or working processes, to highlight the publication value of this manuscript. Finally, provide experimental verification of the effectiveness of this solution in addressing challenges.
Response 1: In the Abstract, we described the unique system design and operational characteristics, along with the experimental validation results.
Comments 2: The introduction can be improved. Firstly, the authors should reorganize the content around the challenges, introduce the efforts and gaps of international peers in addressing the relevant challenges, and then introduce the research motivation of this manuscript. Secondly, the citation and annotation of references are unreasonable, reflecting that the authors have not carefully studied the work of international peers and have not grasped the research progress. Thirdly, at the end of the introduction, the authors should list the contributions of this manuscript one by one, including the key technical details used to address the challenges, i.e., the unique system structure or working process, and experimental design. Meanwhile, ensure that these contributions are consistent with the abstract. Finally, use a paragraph to introduce the organizational structure of the entire manuscript.
Response 2: The Introduction was restructured around the key challenges and supplemented with a review of previous and related studies.
Comments 3: Chapter 2 can be improved. From the beginning, the authors were just experimental setups, and it is completely unclear how they solved the challenges or how they deeply thought about the system structure and working methods. This is regrettable and greatly reduces the publishing value of this manuscript. Suggest that authors provide system architecture diagrams and method flowcharts, and provide detailed explanations of the work process.
Response 3: The Method section begins by proposing solutions to the challenges outlined in the Introduction and is supplemented with a system diagram to enhance the explanation.
Comments 4: Chapters 3 can be improved. Chapter 3 lacks credibility. Firstly, the authors lack sufficient experimental information, including detailed information on the dataset, software and hardware platform data, algorithm parameter settings, and comparative analysis of relevant work from international peers. Secondly, the relevant performance indicators and calculation methods are completely unclear, with only one equation in the entire manuscript, and the accuracy claimed by the authors has no calculation method at all. Finally, the experiment and data need further verification. For example, what is the difference between slow lying down and slow falling? For another example, in section 3.4, line 420, there are two data points, 90% and 98.66%. However, in chapter 3.5, line 438, there are also two data points, 98.66% and 95%. Why are the two not consistent, and how do international peers believe in this accuracy of 90% or 95%?
Response 4: We have described the method used to calculate the model's accuracy.
In this system, there is no separate category for 'slow falling'; instead, it is designed to distinguish between slow lying-down actions and actual falls.
Initially, the accuracy was presented as 90% based on one misclassification out of 10 fall attempts, and 95% based on one misclassification out of a total of 20 trials, including 10 slow lying-down actions and 10 falls.
However, recognizing that such a description may lead to misunderstandings, we have revised the explanation to make it clearer.
Comments 5: Suggest a separate chapter for the conclusion, Chapter 4. The conclusion should summarize whether the entire manuscript effectively addresses the challenges and whether there has been progress compared to international peers. Additionally, it is necessary to identify the shortcomings of this manuscript and future targeted research directions.
Response 5: In response to the reviewer’s feedback, we revised the Discussion and Conclusion sections to improve the clarity of result interpretation and enhance the overall completeness of the manuscript.
Comments 6: The references can be improved. I hope to increase more SOTA work for discussion and comparative experiments, especially in the past two years.
Response 6: To provide a more comprehensive overview of the research field, additional references to related studies have been included in the manuscript.

Reviewer 3 Report
Comments and Suggestions for Authors
The authors proposed a non-contact fall detection system for elderly safety by using a 4D radar. They apply a CNN model to clasify human postures as standing, sitting and lying. They provide a web-based dashboard to visualize the 3D avatars and receive alerts in case of a fall. Although the proposed CNN model achieves high accuracy, the manuscript lacks methodological rigor and offers limited novelty to the literature.
A few additional comments:
- Font inconsistency in lines 130–132 – clarify whether “four directions” implies four separate radars
- Vague description of “various environments and under various conditions”
- Omission of hardware specifications for data acquisition and AI model training
- Lack of details on batch size, learning rate, and training vs. validation loss
- Absence of comparative evaluation against other state-of-the-art models
- Very limited sample size of only ten subjects
The authors are invited to improve their work by reviewing other relevant research, such as:
- for Radar Tensor-based human pose (RT-Pose) dataset and an open-source benchmarking framework: https://link.springer.com/chapter/10.1007/978-3-031-73036-8_7
- https://ieeexplore.ieee.org/abstract/document/9455116
- https://ieeexplore.ieee.org/abstract/document/10459050
Author Response
Comments 1: Font inconsistency in lines 130–132 – clarify whether “four directions” implies four separate radars
Response 1: To enhance the consistency of the manuscript, font inconsistencies were corrected, and in Section 4.3 (Fall Detection Monitoring System), the addition of a virtual camera within the Unity Engine was clearly described in the text.
Comments 2: Vague description of “various environments and under various conditions”
Response 2: To provide a clearer explanation of the various environments and conditions under which the experiments were conducted, revisions were made to the Discussions and Conclusions sections of the manuscript.
Comments 3: Omission of hardware specifications for data acquisition and AI model training
Response 3: To enhance the completeness of the manuscript, the hardware specifications used for data collection were added to Section 3.1 (Experimental Setup).
Comments 4: Lack of details on batch size, learning rate, and training vs. validation loss
Response 4: To improve the completeness of the manuscript, detailed descriptions of the batch size, learning rate, number of epochs, and the training and validation procedures were added to Section 3.4 (Design of the Artificial Intelligence Model).
Comments 5: Very limited sample size of only ten subjects
Response 5: The limited number of participants is recognized as a limitation of this study, and in future research, we plan to recruit additional subjects to expand the participant pool.

Round 2
Reviewer 1 Report
Comments and Suggestions for Authors
The authors have addressed most of the reviewer's concerns. There are a few suggestions from the reviewer.
- The idea of temporal variation for long-term monitoring should be correctly cited.
- Future work part should be moved to discussion and the conclusion section should summarize the entire work in the high level.
- Is the length of the abstract longer than the journal's requirement?
Author Response
comments 1 : The idea of temporal variation for long-term monitoring should be correctly cited.
Response 1 : In response to the reviewer’s comment, we have added appropriate citations in the Discussions section to clarify the concept of temporal variability in long-term monitoring. This addition aims to enhance the readers’ understanding of this important aspect.
comments 2 : Future work part should be moved to discussion and the conclusion section should summarize the entire work in the high level.
Response 2 : In accordance with the reviewer’s suggestion, we have moved the content related to future work to the Discussion section. The Conclusion section has been revised to provide a high-level summary of the entire study.
comments 3 : Is the length of the abstract longer than the journal's requirement?
Response 3 : In response to the reviewer’s comment, we have reviewed the abstract length to ensure it complies with the journal’s requirements and revised it accordingly.

Reviewer 2 Report
Comments and Suggestions for Authors
Dear authors,
Thank you for carefully revising the manuscript and responding to the review comments one by one. The quality of the manuscript has significantly improved compared to the original version. However, there are still some flaws that need to be addressed before formal publication.
- The phrase 'Validation Study Based on' in the title can be removed.
- In the keywords, 'Fall detection' should be placed first.
- The abstract can be appropriately compressed, leaving only the most important challenges, unique technical features, and contributions.
- It is best to list the contributions at the end of the introduction paragraph by paragraph, that is, one small paragraph corresponds to one contribution, and it should be consistent with the revised abstract. At the end of the introduction, add another paragraph to introduce the organizational structure of the entire manuscript. In addition, the citation of references should be targeted and interwoven, rather than the current situation, such as the end of the first paragraph [1], the end of the second paragraph [2-5], and the end of the third paragraph [6-7]. Such citation is not serious. If published in this form, international peers will have negative opinions on this manuscript and this journal.
- Chapter 2 needs to expand its content.
- Change the title of Chapter 3.1 to Technical Architecture. There are too few equations in the entire manuscript, only one. Please further supplement, such as the CNN calculation process.
- Chapter 4 lacks credibility, how is accuracy calculated? Please supplement the equation.
- Please discuss Chapter 5 in conjunction with the previous figures, tables, and specific data.
Author Response
comments 1 : The phrase 'Validation Study Based on' in the title can be removed.
Response 1 : In accordance with the reviewer’s suggestion, we have revised the manuscript title by removing the phrase “Validation Study Based on” to enhance clarity and conciseness.
comments 2 : In the keywords, 'Fall detection' should be placed first.
Response 2 : As suggested by the reviewer, we have revised the Keywords section by placing “Fall detection” as the first keyword.
comments 3 : The abstract can be appropriately compressed, leaving only the most important challenges, unique technical features, and contributions.
Response 3 : In accordance with the reviewer’s recommendation, we have revised the abstract by condensing its content to include only the most critical problem, the unique technical features of our approach, and the main contributions of the study.
comments 4 : It is best to list the contributions at the end of the introduction paragraph by paragraph, that is, one small paragraph corresponds to one contribution, and it should be consistent with the revised abstract. At the end of the introduction, add another paragraph to introduce the organizational structure of the entire manuscript. In addition, the citation of references should be targeted and interwoven, rather than the current situation, such as the end of the first paragraph [1], the end of the second paragraph [2-5], and the end of the third paragraph [6-7]. Such citation is not serious. If published in this form, international peers will have negative opinions on this manuscript and this journal.
Response 4 : In response to the reviewer’s valuable suggestions, we have revised the citation style in the Introduction section to present references in a more contextually integrated and precise manner. Furthermore, we restructured the final part of the Introduction to list each contribution in separate paragraphs and added a description of the organizational structure of the paper.
comments 5 : Chapter 2 needs to expand its content.
Response 5 : In response to the reviewer’s comment, we have expanded the content of Chapter 2 to provide a more comprehensive overview of the research background and related work.
comments 6 : Change the title of Chapter 3.1 to Technical Architecture. There are too few equations in the entire manuscript, only one. Please further supplement, such as the CNN calculation process.
Response 6 : In accordance with the reviewer’s suggestion, we have revised the title of Section 3.1 to “Technical Architecture.” Furthermore, in response to the comment regarding the limited use of equations, we have supplemented the manuscript by adding equations related to the preprocessing steps described in Section 3 and the accuracy calculation methods presented in Section 4.
comments 7 : Chapter 4 lacks credibility, how is accuracy calculated? Please supplement the equation.
Response 7 : In response to the reviewer’s comment regarding the reliability of Chapter 4, we have enhanced the explanation of the accuracy evaluation method and included the explicit equation used to calculate accuracy.
comments 8 : Please discuss Chapter 5 in conjunction with the previous figures, tables, and specific data.
Response 8 : In accordance with the reviewer’s suggestion, we have revised the Discussion section (Chapter 5) to more closely link the discussion with the previously presented figures, tables, and numerical results.

Reviewer 3 Report
Comments and Suggestions for Authors
The authors have improved the paper by providing more information about their research design and discussions.
The paper should still be improved by clarifying how good are their results in comparison with other solutions.
What is the rationale of using the Retina-4sn radar when this device already provides fall detection and prediction, as well as detecting the three human postures of standing, sitting, and lying down? Moreover, the device could even measure vital signs, such as breathing and pulse at a high accuracy.
Perhaps the virtual environment in Unity could be linked to concepts related to the Metaverse?
Author Response
comments 1 : The paper should still be improved by clarifying how good are their results in comparison with other solutions.
Response 1 : In response to the reviewer’s comment, we have enhanced the manuscript by providing a clearer comparison with existing solutions. This comparison has been incorporated into both Section 2 and the Conclusion to better highlight the advantages of our proposed approach.
comments 2 : What is the rationale of using the Retina-4sn radar when this device already provides fall detection and prediction, as well as detecting the three human postures of standing, sitting, and lying down? Moreover, the device could even measure vital signs, such as breathing and pulse at a high accuracy.
Response 2 : As the reviewer correctly pointed out, the Retina-4sn radar includes built-in functionalities such as posture recognition and biosignal measurement. However, in this study, we did not utilize any of the device’s built-in features. Instead, we employed independent preprocessing techniques and a custom-designed AI model to reduce computational complexity and enable a lightweight architecture suitable for real-time applications. The fall detection algorithm was also developed independently, as proposed in this paper, to effectively distinguish between lying down and falling. Furthermore, we used a version of the Retina-4sn radar that does not include biosignal detection capabilities such as respiration or pulse monitoring.
comments 3 : Perhaps the virtual environment in Unity could be linked to concepts related to the Metaverse?
Response 3 : In line with the reviewer’s suggestion, we acknowledge that the Unity-based virtual environment used in this study holds potential for future expansion toward digital twin applications and integration with metaverse concepts. The avatar-based visualization not only enhances real-time monitoring but also offers an intuitive representation of the user’s state, indicating the possibility of broader applications in digital healthcare systems.
